# Hardening by Transformation and Cold Working in a Hadfield Steel Cone Crusher Liner

Rodrigo Allende-Seco [1,*], Alfredo Artigas [1,2], Héctor Bruna [1], Linton Carvajal [1], Alberto Monsalve [1] and María Florencia Sklate-Boja [3]

1   Departamento de Ingeniería Metalúrgica, Universidad de Santiago de Chile, Santiago 9160000, Chile; alfredo.artigas@usach.cl (A.A.); hector.bruna@usach.cl (H.B.); linton.carvajal@usach.cl (L.C.); alberto.monsalve@usach.cl (A.M.)
2   Laboratorio SIMET-USACH, Santiago 9170124, Chile
3   Instituto de Física de Rosario, UNR-CONICET, Rosario 2000, Argentina; sklateboja@ifir-conicet.gov.ar
*   Correspondence: rodrigo.allende@usach.cl

**Abstract:** This paper presents the characterization of a secondary cone crusher concave liner made of Hadfield steel used in Chilean mining after crushing copper minerals during all service life. During use, a cone crusher concave liner suffers indentation (cold working) and abrasion; this combination provides the concave with a layer that constantly renews itself, maintaining a surface highly resistant to abrasive wear. The results presented here were obtained using optical microscopy, microhardness test, measuring abrasion using the dry sand/rubber wheel apparatus, and x-ray diffraction peaks analysis through the classic Williamson–Hall method. After analysis of results, two hardened surfaces have been found—one a product of heat treatment and the other due to deformation during use. This work proposes ways to explain them; the first one uses a thermodynamic model to calculate stacking fault energy, and the second compares the liner with cold-rolled samples.

**Keywords:** austenitic steel; stacking fault energy; hardening; phase transformation; cold working



## 1. Introduction

Cone crushers are used by the mining industry and its purpose is to reduce particle size of rock materials or to liberate valuable minerals from ores. The 200 mm material (that comes from primary crushing) is fed through the top of the equipment and is repeatedly compressed (fractured) until a product of approximately 75 mm (that goes towards tertiary crushing) is obtained that is expelled from the bottom.

The original austenitic manganese steel, Hadfield steel, was patented by Robert Hadfield in 1882 and contained C and Mn in a ratio of approximately 1/10 [1]. Several modifications have been proposed to increase its mechanical or corrosion resistance. The main characteristic of these steels is their remarkable capacity to be hardened by deformation while keeping toughness; this is the reason why Hadfield steels are ideal for uses that require wear resistance and work hardening capacities, such as in the mining industry where, for instance, they are used as cone crusher liners which are subjected to pure indentation (work hardening) and abrasion (wear) [2].

Several studies have been carried out to establish the mechanisms involved during the deformation of these materials, explaining the high work hardening. The above depends on the austenite stability, which controls the facility of deformation-induced martensitic transformation of austenite ($\gamma$) into $\varepsilon$ martensite, $\alpha'$ martensite, or mechanical twinning. Several authors [3–6] have studied the relationship between deformation mechanism and stacking fault energy (SFE), determining SFE ranges vs. occurrence of these different mechanisms. For the estimation, it is assumed that SFE is the Gibbs energy required to create a platelet of $\varepsilon$ martensite of a thickness of only two atomic layers. According to Allain et al. [5] and data in the literature [7,8], displacive transformations can occur during plasticity depending on

the SFE. In these articles, it has been commented that for SFE < 18 mJ m$^{-2}$ the ε martensitic transformation occurs, and for 12 mJ m$^{-2}$ < SFE < 35 mJ m$^{-2}$, mechanical twinning takes place.

To estimate SFE variations in this work, the thermodynamic model described by Allain for the FeMnC system [5] was used. The term $\Delta G_{ex}$ was incorporated into the model to include the effect of grain size on SFE in austenitic steels, as studied by Takaki [9] The breakdown of this modified model with the influence of grain size is shown in Figure 1. In addition, the numerical values and equations used for the calculations performed in this work are shown. In those equations, ρ is the molar surface density along {111} planes, $\Delta G^{\gamma \to \varepsilon}$, the molar Gibbs energy of the transformation γ/ε, σ the surface energy of the interface γ/ε, a, the lattice parameter, N, the Avogadro number, R, the gas constant, T, the deformation temperature, $\beta^{\varphi}$, the magnetic moment, $x_i$, the atomic fraction, and d, the grain size in microns.

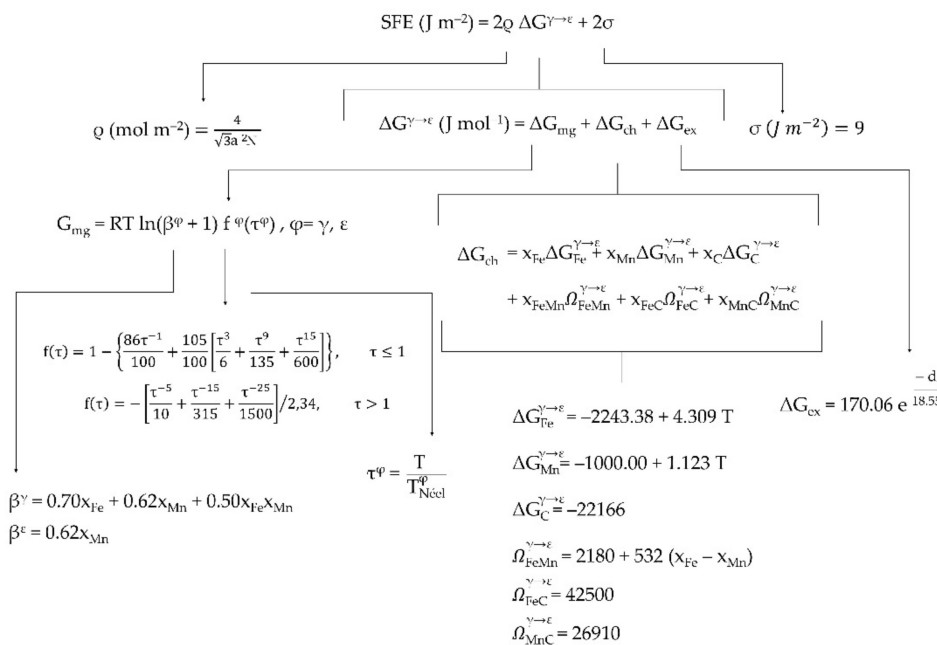

**Figure 1.** Breakdown of the modified thermodynamic model and numeric values used in this work.

Several contributions have been made to the relationships between SFE, microstructure [5,6], and grain size [9], and wear resistance [10,11]. Similarly, the effect of strain rate by impact through shot peeing [12,13], and explosive hardening [14,15] on the microstructure in austenitic steels with 12 mJ m$^{-2}$ < SFE < 35 mJ m$^{-2}$ has been studied. Most of the articles related to crusher liners have been approached from a simulation perspective [16–18]. After use, the crusher liners are recycled, so access to these samples is difficult; therefore, it is not possible to find information on these types of parts during or after use.

This work aims to carry out applied research in engineering to relate the information described in the literature on austenitic manganese steels, with the bases of materials science and engineering, to lay a foundation to advancing the in-depth understanding of the relationship between the structure and the properties of austenitic alloys. This is carried out by interpreting results obtained by optical microscopy, X-ray diffraction, hardness tests, and measurements of resistance to abrasion of a cone crusher liner after its manufacture and use.

## 2. Materials and Methods

A worn secondary cone crusher concave liner (in what follows concave) made of Hadfield steel with a composition (wt.%) of Fe-12.33Mn-1.12C-0.54Si-0.067P-0.015S-1. 43Cr-

0.026Ni-0.014Mo was provided by a mining supply company and used in this work. Figure 2 shows a cone crusher diagram, where some features, such as feed opening, open size setting (OSS), and close size setting (CSS), are shown. Additionally, three zones of interest are highlighted: (zone 1) bulk, (zone 2) outer surface, and (zone 3) crushing surface. For the results and discussions section, the colors used to show the three zones of interest will be used. Before assembly in a crusher, the concave liner was heated at 1100 °C for 6 h, followed by water quenching to dissolve carbides and avoid precipitation during cooling, to obtain a completely austenitic structure. Once assembled, the concave (250 mm thick) was used for crushing copper ores until it reached its useful life and when it lost 60% of its initial weight, its final thickness was approximately 100 mm.

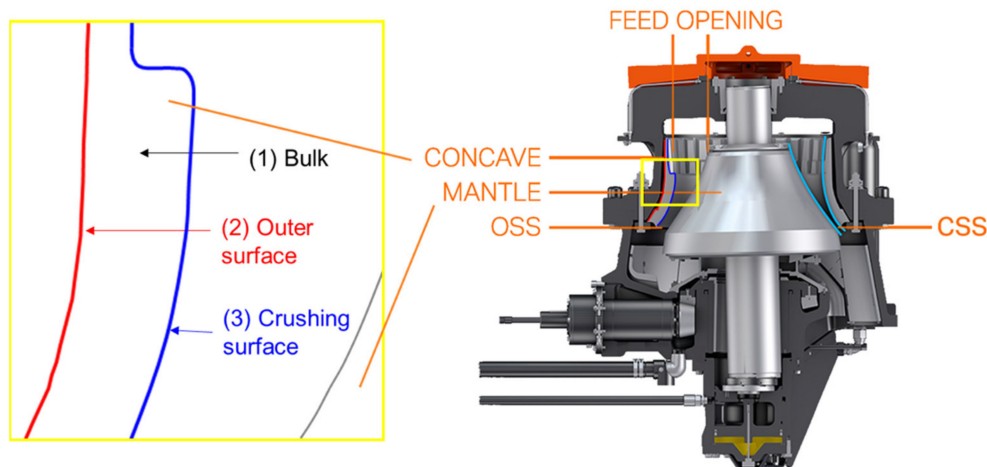

**Figure 2.** Diagram of a cone crusher showing a concave section.

Before expanding details on the experimental techniques used, it is worth mentioning that in the sample studied in this work, the microstructure of zones 1 and 2 is not modified, since they are not subjected to plastic deformation or wear during the operation of the crusher. Before the operation, it is expected that the surface microstructure of the concave would have been like that of zone 2. Once the ore crushing begins, the face exposed to the crushing chamber (zone 3) is continuously removed by abrasion and renewed by the indentation of copper ore. The indentation hardens the crushing surface by deformation. This causes an increase in resistance to abrasive wear; both processes occur until the end of its useful life, when original concave loses 60% of its initial mass. Therefore, the microstructure of the crushing surface is expected to remain in constant renewal.

For characterization purposes, samples from the three zones were cut with a wire electrical discharge machine (EDM) (Novick, Beijing, China) from the concave. Vickers hardness profiles of the samples were obtained using a Struers Vickers hardness tester using 100 gf test force (HV0.1 in accordance with ASTM E92 [19]). Metallographic examinations were conducted by optical microscopy (OM) (Olympus, Tokyo, Japan) after the samples had been grinded in SiC paper and polished with polycrystalline diamond suspension and then etched using Nital's etchant. Phases present were identified by comparison with data in the literature and grain size was measured using the linear intersection method described in ASTM E112 [20] and ASTM E1181 [21].

For phases identification, calculations of crystallite sizes and microstrains, from the three zones, were obtained from x-ray diffraction peak analysis using the classical Williamson–Hall [22] method. For the preparation of the samples corresponding to zones 2 and 3, the normal faces have been prepared in the same way in which the samples for OM were prepared. The diffraction profiles have been obtained at 400 microns from the surface in both cases. Moreover, 400 microns corresponds to the thickness lost during sample preparation to obtain a flat surface of 400 mm$^2$ to avoid peaks broadening caused

by X-ray divergence. Three X-ray diffraction profiles were measured in each of the areas of interest.

X-ray diffraction profiles were measured using a Rigaku Miniflex 400 diffractometer fitted with a CrKα tube (Toshiba, Tokyo, Japan). Measurements were made between 60 < 2θ <135 degrees at a scanning speed of 1 degree per minute. To record the scattered x-rays, a high-speed Rigaku D/TeX Ultra detector (Rigaku, Tokyo, Japan) has been used. To minimize instrumental and experimental errors, the diffractometer (Rigaku, Tokyo, Japan) was calibrated according to the supplier's recommendations, then drift and broadening of the instrument's own peaks were calculated using a $LaB_6$ standard (NIST 660C). Then, the measurements of the samples were corrected assuming a Lorentzian (Cauchy) profile. Using the same profiles, the phases present in the samples were identified.

The abrasion resistance of the three zones was measured using a dry sand/rubber wheel apparatus (Own development, Santiago, Chile) (ASTM G65 [23]). Calibration of the apparatus was carried out using SS 316 standard reference specimens with a hardness of 80 HRB. Weight losses were determined after 718 m of linear wear. For additional details on the methodology used to evaluate abrasive wear resistance, see ASTM G65 [23].

For comparison and to visualize a relationship between deformation and resistance to abrasive wear, several samples of $75 \times 25 \times 10$ mm$^3$ were cut from the bulk of the concave and then deformed by cold rolling to 0.05, 0.11, 0.16 and 0.22 (expressed as natural or logarithmic strain). Lamination was performed on an electrical rolling mill (Joliot, Paris, France) with 150 mm diameter rollers, 190 mm width and a rotation speed of 18 rpm. Metallographic examinations and abrasion resistance measurements were made to deformed samples.

## 3. Results and Discussion

In this section, results and discussions obtained from the experiments described above will be presented. Figure 3 shows images obtained by light microscopy of the three areas of interest.

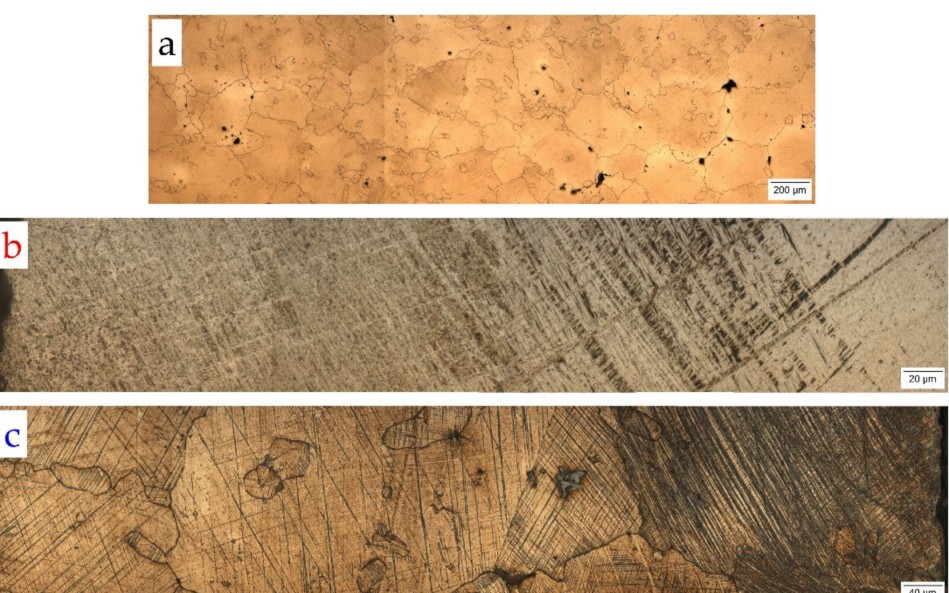

**Figure 3.** Optical micrographs of (**a**) bulk zone (zone 1); (**b**) outer surface (zone 2); (**c**) crushing surface (zone 3).

The original bulk's microstructure (zone 1) is composed of austenitic grains with a duplex size. It has been calculated by the linear intersection method that 15 percent has a size of $47 \pm 5$ μm and the remaining percentage $512 \pm 63$ μm, without the presence of deformation bands or carbides (Figure 3a); its hardness is $278 \pm 5$ HV 0.1, and after

718 m of linear abrasion, it experienced a weight loss equivalent to 411.9 ± 1.6 mg. In zone 2 (Figure 3b), it is possible to observe ε-martensite, of which the quantity decreases from the outer surface, and at a distance greater than 600 microns, it is possible to observe thermal twins. In zone 3 (Figure 3c), austenite is observed with a large presence of twins on the crushing surface, which diminishes towards the interior. In zones 2 and 3, superficial hardening, as shown in Figure 4, is caused by changes in the microstructure. Hardness decreases gradually from both surfaces towards the bulk. To facilitate understanding, zones 2 and 3 will be discussed separately.

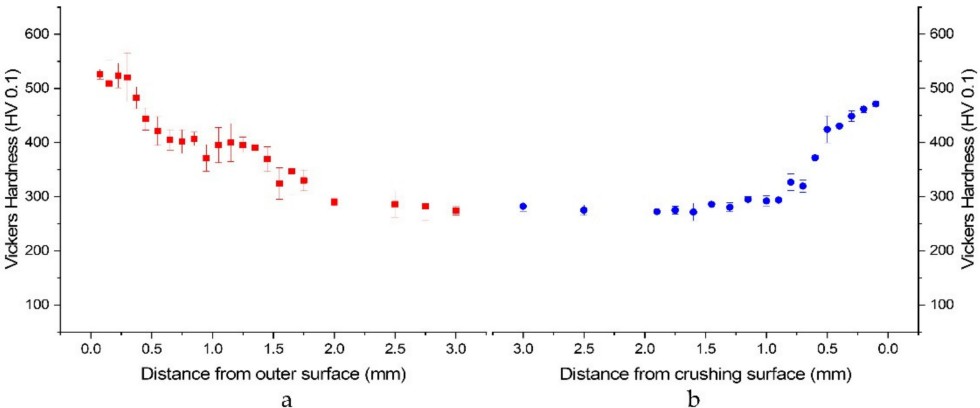

**Figure 4.** Variation in hardness with (**a**) distance from outer surface and (**b**) distance from crushing surface after use from worn concave of Hadfield steel.

Figure 3b shows martensite whose content decreases from left to right being replaced by austenite. Similarly, in Figure 3c, austenite can be observed, with many twins decreasing from right to left. The morphologies are consistent with what is shown in the chapter "Austenitic Manganese Steel Castings" in volume 9 of the ASM Metals Handbook series. [24].

### 3.1. Outer Surface (Zone 2)

Zone 2 represents the surface exposed to the atmosphere during the heat treatment. The microstructure is composed of ε-martensite (hexagonal) on the outside with an increase in the amount of austenite towards the inside. Figure 5 shows an X-ray diffraction profile at ~400 microns under the quenched surface in which it was possible to identify the described phases. In general, in FCC phases, transformation can be produced by a decrease in the stacking fault energy [4,6,7]; in this case, it occurs due to the decrease in the carbon content at the surface associated with heat treatment during the manufacturing process.

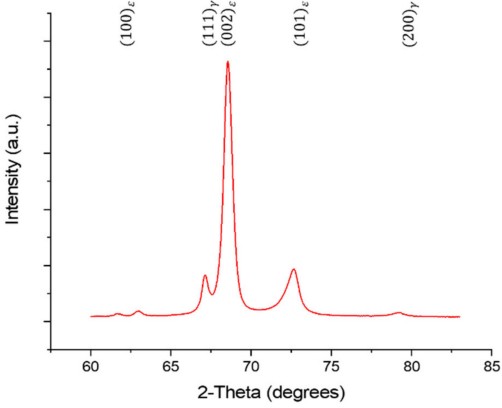

**Figure 5.** X-ray diffraction profile obtained at ~400 μm from the quenched surface.

To illustrate the effect of the decrease in carbon content on the microstructure, the following aspects were considered: (1) The heat treatment prior to use consisted of heating at 1100 °C for 6 h in an oxidizing atmosphere, then quenched in water; (2) the carbon content evolves during heat treatment according to Fick's second law for diffusion; (3) the functional relationship between Fe, Mn, C and SFE contents is as described by Allain et al. [5].

For the calculation of C content vs. distance from quenched surface, Fick's second Law [25] was solved, assuming a semi-infinite system with a carbon concentration (by weight) in the atmosphere equal to zero, the bulk concentration equal to that of the alloy studied, at a temperature equal to 1100 °C for 21,600 s, and a diffusion coefficient of $7.18 \cdot 10^{-12}$ m$^2$/s according to the work of Král et al. [26]. Then, from these results, the relationship between SFE and distance from the quenched surface was calculated at room temperature, using the data shown in Figure 1; the effect of grain size on SFE ($\Delta G_{ex}$ in Figure 1) was neglected due to its low contribution in this case; if the term $\Delta G_{ex}$ is calculated using the grain size of approximately 500 microns, this term has a weighting close to zero. The SFE profile obtained is shown in Figure 6.

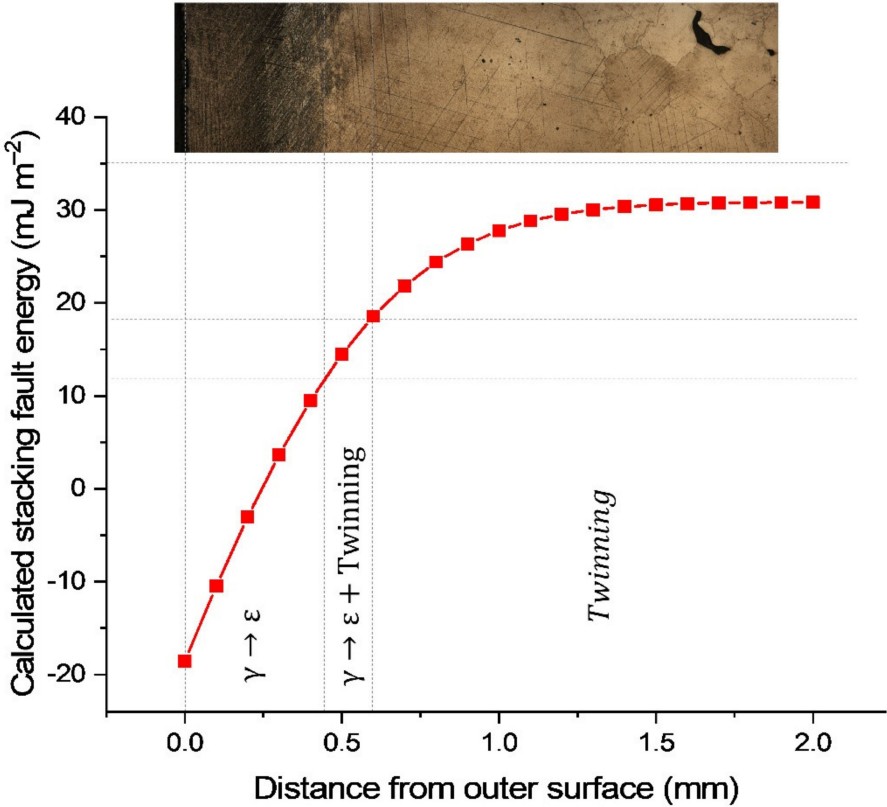

**Figure 6.** SFE profile, relationship between SFE-Deformation mechanism and microstructure of the decarburized Hadfield steel.

Ranges of the SFE for phase transformation and deformation twinning reported for Allain et al. [5] are also shown. Additionally, a metallography of the outer surface is superimposed to show the relationship between the surface microstructure after heat treatment and the calculations performed. The scales on the micrograph and the distance axis are equivalent. Thus, it is shown that the studied Hadfield steel is hardened by transformation from austenite to ε-martensite due to the SFE profile that generates surface decarburization. Although a surface hardening is observed in zone 2, this is not desirable. Kolokolsev et al. [10] have reported that the decrease in SFE decreases resistance to wear.; to minimize decarburization during heat treatment, additions of 2.5 percent chromium are common [11]. The decarburized layer is negligible (approximately 1 mm) with respect to

the total thickness lost during use (150 mm), so decarburization is not important in this case. To solve issues connected to the decarburization of the surface-near region, it would be worth considering a certain oversize of the part for heat treatment, which could be removed by grinding afterwards. First, however, it would be necessary to evaluate if this is possible considering the increase in cost due to material loss and the grinding process.

### 3.2. Crushing Surface (Zone 3)

Now, the effect of use in a cone crusher on the microstructure of a Hadfield steel and its relationship with abrasion resistance will be studied. For this, the following aspects will be considered: (1) Zone 3 is the surface of the concave exposed to the crushing chamber once the concave has completed its useful life. (2) The microstructure (Figure 3c) comprises austenitic grains with different numbers of twins, which decrease from the crushing surface. This means that different amounts of deformation develop from the surface towards the interior. (3) During use it is not possible to measure stresses and strains near the surface, (4) an estimation of the macroscopic deformation on the crushing surface will be made from the relationship between hardness and rolling strain.

Figure 7 shows the results obtained for cold rolled samples obtained from zone 1. Additionally, the surface hardness of zone 3 (crushing surface) and its resistance to abrasion are shown in blue. The results show that the increase in deformation causes an increase in hardness, and this generates an increase in resistance to abrasion (decrease in weight loss). The standard deviation value at 0.05 rolling strain is high, since one of the tested samples had a greater number of internal cracks; this caused an increase in weight loss during the abrasive wear test. Hardness and weight loss are similar in the crushing surface and the sample with rolling strain equal to 0.22. This shows that the concave experiences an increase in its resistance to abrasion (increase in hardness) due to the change in the microstructure produced by use. A model to understand the hardening mode has been proposed by Liu et al. [14]. The hardening caused by the microstructural change at room temperature has been attributed to twinning in this type of materials in a similar way to that described for the Hall–Petch effect, because mechanical twinning causes a decrease in dislocation mean free path [27].

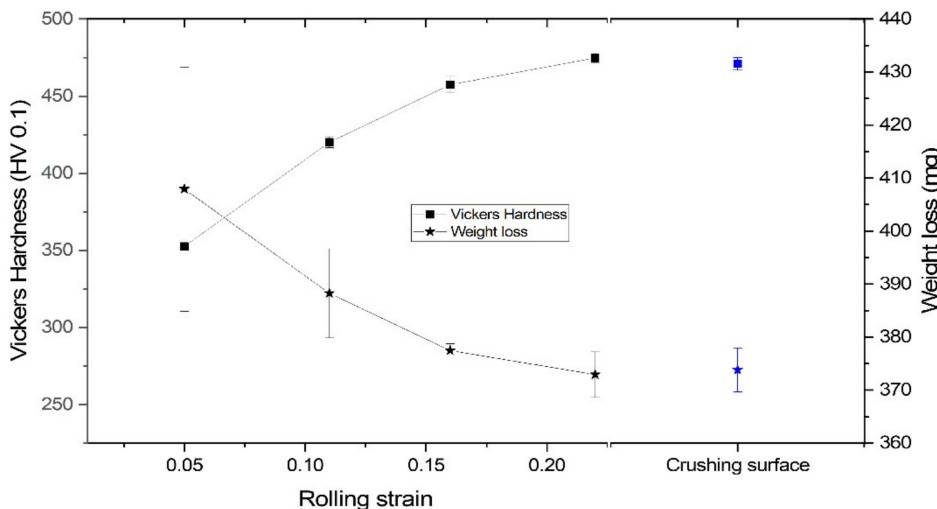

**Figure 7.** Relationship between rolling strain vs. hardness and weight loss (according to ASTM G65) in a Hadfield steel.

As shown in the work of Liu et al. [14], it will be assumed that the rolling strain that results in an increase of the same hardness as during use as a cone crusher is considered as the estimated deformation. Using both the relationship between hardness and rolling strain shown in Figure 7 and the hardness profile from the crushing surface in Figure 4b, Figure 8 has been obtained; this shows the variation in estimated strain with distance

from the work hardening surface of the concave after use. The assumed absolute strain in crushing surface, that is, the distance that the crushing surface recedes because of the mineral indentation against was determined by the area under the curve in Figure 8 to be 100 microns. The above means that the concave showed a low macroscopic deformation during its use. If the calculated estimated strain value is compared with others, such as the case of shot peening [12,13] or explosion hardening [14,15], we can comment that the size of the surface hardening is greater than that obtained by shot peening and less than in explosion hardening. The situation described above is like the hardening suffered by the surface of an alloy that has been indented by a steel ball as in the Brinell hardness test. Due to the indentation; then, the surface recedes and the microstructure of the region close to the contact with the steel ball and the alloy is modified, causing strain hardening.

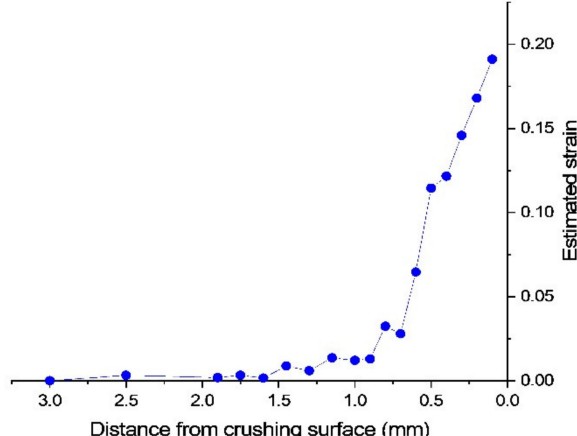

**Figure 8.** Variation in assumed strain with distance from the work hardened surface of the concave.

Indentation in a secondary cone crusher occurs between 5 and 8 times per second, if we consider how it operates [28,29]; this is a higher speed than that caused by rolling using the conditions described above. Therefore, the deformation estimated by Figure 8 is underestimated if the negative strain rate dependence of Hadfield steel is considered [30]. It is then possible to say that since the strain speed is high enough and therefore, according to the assumptions made, the assumed absolute deformation would be at least 100 microns when indentation occurs on the crushing surface.

Another way to show that the deformation during use is greater than that estimated using rolling is by analyzing X-ray diffraction peaks. The Williamson–Hall method [18] considers that peak broadening is attributed to deformation and crystallite size; for this reason, the effects of alloying elements, stacking failures, twins, or others have not been considered. Figure 9 shows (a) peak broadening by deformation for planes (111) and (b) the conventional Williamson–Hall plot. In Figure 9, the terms $\theta_{exp}$, $\theta_{(111)}$, $\beta_r$ and $\theta$ correspond to experimental data obtained from X-ray diffraction experiments, Bragg angle calculated assuming a Lorentzian distribution of the (111) peak, full width at half maximum (FWHM) and Bragg angle, respectively. The samples called rolled and crushing surface for Figure 9 correspond to the one that was deformed 0.22 by rolling and the other at 400 microns below the crushing surface, respectively. In Figure 9a, peak broadening is greater in a sample obtained 400 microns towards the interior of the crushing surface than for the rolled sample; this could mean that it is more strained. This last information can be verified by analyzing Figure 9b, where a greater slope and displacement towards higher abscissa values are observed in the conventional Williamson–Hall plots [18], and where it is also possible noticing how the microstrain increases as the crystallite size decreases. Average microstrain (calculated by slope) and crystallite sizes (calculated by y-intercept) were obtained from the conventional Williamson–Hall plot and the density of dislocations using Equation (1) [31]:

$$\rho \cdot L \cdot b = 2 \cdot (3 \cdot \sigma^2)^{1/2}, \tag{1}$$

where ρ is the dislocation density, L is crystallite size, σ is the average microstrain, and b is Burgers vector, which is equal to $a \cdot 2^{-1/2}$ for the <110> direction of austenite; a is the lattice parameter calculated from the X-ray diffraction profiles using Bragg's law [22]. The purpose of performing these calculations is for explanatory purposes. The summary of the analysis of X-ray diffraction peaks is shown in Table 1.

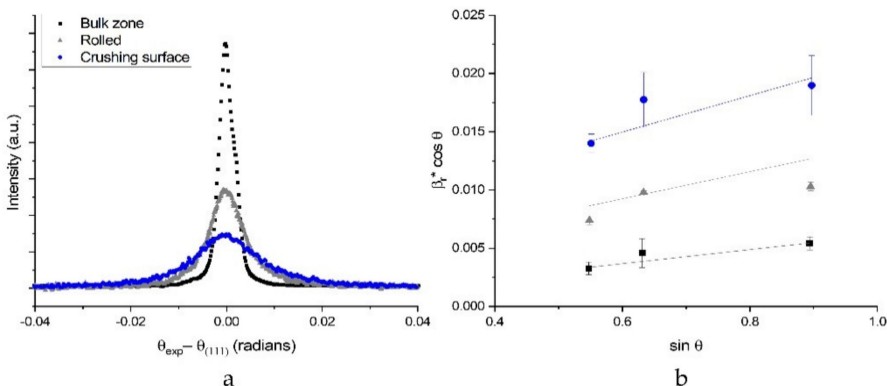

**Figure 9.** X-ray diffraction peak analysis. Effect of strain vs. (**a**) (111) peak broadening and (**b**) the conventional Williamson–Hall plot.

**Table 1.** Summary of X-ray diffraction peak analysis.

| Sample Identification | Average Microstrain $10^{-5}$ | Crystallite Size nm | Dislocation Density $10^9$ cm$^{-2}$ |
|---|---|---|---|
| Bulk zone | 439 ± 258 | 106 ± 46 | 63 ± 3 |
| Rolled 22 percent | 545 ± 138 | 50 ± 17 | 151 ± 15 |
| Crushing Surface under 200 microns | 955 ± 326 | 24 ± 3 | 527 ± 133 |

## 4. Conclusions

In summary, after heat treatment, the microstructure of the surface of a cone crusher has been transformed to ε-martensite by decarburization in an oxidizing atmosphere. The surface layer is constantly subjected to deformation (due to mineral indentation that causes the relative movement of the concave with respect to the mantle) and abrasion (due to mineral drag during size reduction). This causes surface wear and continuous renewal of the hardened layer with greater resistance to abrasion than the bulk material, with a small macroscopic strain, on the order of hundreds of microns. This is assumed, since the results obtained by analyzing diffraction peaks are in the same order of magnitude, even when they are different for the sample deformed in the crusher than for the deformed by rolling.

After analyzing a cone crusher liner used for crushing copper ores to the end of its useful life, an explanation for the surface hardening observed in decarburized Hadfield steels has been proposed, using a thermodynamic model for stacking fault energy calculation. Furthermore, it has been shown that the small amount of macroscopic deformation of the order of hundreds of microns that occurs during the operation of the crusher is sufficient to increase the hardness of the crushing surface, also increasing the resistance to abrasive wear. This hardening is produced by a decrease in dislocation mean free path like that described for the Hall–Petch effect.

Several contributions have been made to the relationships between SFE, microstructure, and grain size, and wear resistance. Similarly, the effect of strain rate by impact through shot peening, and explosive hardening on the microstructure in austenitic steels with 12 mJ m$^{-2}$ < SFE < 35 mJ m$^{-2}$ has been studied. Most of the articles related to crusher liners have been approached from a simulation perspective. After use, the crusher liners are recycled, so access to these samples is difficult; therefore, it is not possible to find information on these types of parts, during or after use.

This work has carried out applied research in engineering to relate the information described in the literature on austenitic manganese steels with the bases of materials science and engineering, to lay a foundation to advancing the in-depth understanding of the relationship between the structure and the properties of austenitic alloys. This was carried out by interpreting results obtained by optical microscopy, X-ray diffraction, hardness tests, and measurements of resistance to abrasion of a cone crusher liner after its manufacture and use.

**Author Contributions:** Conceptualization, R.A.-S.; Data curation, R.A.-S.; Formal analysis, R.A.-S.; Funding acquisition, R.A.-S., A.A. and A.M.; Investigation, R.A.-S.; Methodology, R.A.-S. and A.A.; Project administration, R.A.-S. and A.A.; Resources, A.A. and A.M.; Software, R.A.-S.; Supervision, A.A. and A.M.; Validation, R.A.-S., A.A. and A.M.; Visualization, R.A.-S. and M.F.S.-B.; Writing—original draft, R.A.-S.; Writing—review and editing, A.A., H.B., L.C., A.M. and M.F.S.-B. All authors have read and agreed to the published version of the manuscript.

**Funding:** This research was funded by Dirección de Investigación Científica y Tecnológica (DICYT) de la Universidad de Santiago de Chile grant number 051914AA and by the Agencia Nacional de Investigación y Desarrollo (ANID) ANID-PFCHA/Doctorado Nacional/2017-21170167.

**Institutional Review Board Statement:** Not applicable.

**Informed Consent Statement:** Not applicable.

**Data Availability Statement:** Data are contained within the article or supplementary material. The data presented in this study are available in Ph. D. Thesis of Rodrigo Allende entitled "Influence of Stacking Fault Energy on Strain Hardening and Abrasion Resistance in Fe22MnxC Cast Steels", USACH 2021, Santiago, Chile.

**Acknowledgments:** Rodrigo Allende thanks the Agencia Nacional de Investigación y Desarrollo, ANID-PFCHA/Doctorado Nacional/2017-21170167 for financing his graduate studies.

**Conflicts of Interest:** The authors declare no conflict of interest.

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
