# Peer review of "Hardening by Transformation and Cold Working in a Hadfield Steel Cone Crusher Liner"

_metals, doi:10.3390/met11060961_

Round 1

Reviewer 1 Report

In this work, the authors described the mechanisms which result in surface hardening of the Hadfield steel cone crusher liner. The manuscript is appropriate for publication in Metals with major revision. The authors should revise their manuscript according to the following remarks:

Section 3: The authors estimate the macroscopic deformation on the crushing surface from the relationship between hardness and rolling strain. Please give justification (in the manuscript), as rolling will result in different microstructure and texture.

Fig.7, The std. value at 0.05 rolling strain is very high. Please discuss this in the manuscript.

Line 239: Peak broadening was associated with deformation. I wonder how the authors distinguished broadening caused by carbon content in the matrix and deformation. Please explain in the manuscript.

Line 240. Please include a reference to the Williamson-Hall method.

Line 251: The mean size and strain were calculated from the Williamson-Hall plot. The authors cited the work of Louidi et al., which employed the Warren-Averbach method, please correct. Also, the plot display parabolic characteristics; I wonder how the authors calculated size and strain, as in the conventional Williamson-Hall method, these values are commonly extracted using a linear fitting. Please discuss in the manuscript.

Author Response

To see the responses to your comments, please download the attached file. 

Reviewer 2 Report

Dear Authors,

Please, find below several suggestions.

The main concern I have about your paper is the poor quality of the images. The metallographic characterisation is insufficient and please, add better images.

Please, report also the number of measurements carried on. Are the RX data only one shoot per zone? In few word, try to give robustness to Table 1 data.

The dislocation density has been reported and it is very important to explain the method, the model adopted and discuss why you have decided to adopt this simplified estimation. No possibility to carry on TEM observation? Nevertheless, I agree with your conclusion.

Line 30-31: these historical notes are interesting, but a reference is necessary., otherwise delete the two lines.

Figure 3: Quality of Fig.3 b and c are not sufficient to see the microstructure, please, improve it.

Figure 4: the two metallographic images at the two lower sides are illegible.

Line 219: Fig.8 change in “…Figure 8”

Line 254: report the equation utilised not only the reference.

Author Response

(The authors gave the same response as above.)

Reviewer 3 Report

This paper is on characterization of a secondary cone crusher concave liner made of a steel.

It was based on experimental observation during all the service life of the liner. However, it looks like a kind of technical report.

It is thus a good research work for a conference paper but it is not appropriate for a learned journal.

Author Response

Response to Reviewer 3 Comments

This paper is on characterization of a secondary cone crusher concave liner made of a steel.

It was based on experimental observation during all the service life of the liner. However, it looks like a kind of technical report.

It is thus a good research work for a conference paper but it is not appropriate for a learned journal.

Response 1: At the authors' discretion, the article presented meets the expectations of a publishable work in metals, given the scope and objectives that they declare. We agree that it can be interpreted as a technical report. Still, strictly speaking, it corresponds to an investigation in which the object of study has been analyzed from an interpretive paradigm. This has been done to contribute to the learning transfer of elements of the initial training of engineers or scientists of materials.

The text has been modified so that it now reads,

“          Several contributions have been made to the relationships between SFE, microstructure [5-6], and grain size [9], and wear resistance [10,11]. Similarly, the effect of strain rate by impact through shot peeing [12,13], and explosive hardening [14,15] on the micro-structure in austenitic steels with 12 mJ m-2 <SFE <35 mJ m-2 has been studied. Most of the articles related to crusher liners have been approached from a simulation perspective [16-18]. After use, the crusher liners are recycled, so access to these samples is difficult; therefore, it is not possible to find information on these types of parts during or after use.

This work aims to carry out applied research in engineering to relate the information described in the literature on austenitic manganese steels with the bases of materials science and engineering, to lay a foundation to advancing the in-depth understanding of the relationship between the structure and the properties of austenitic alloys. This is carried out by interpreting results obtained by optical microscopy, X-ray diffraction, hardness tests, and measurements of resistance to abrasion of a cone crusher liner after its manufacture and use.”

We sincerely hope that your assessment of the work carried out will be modified according to the above.

Reviewer 4 Report

The considered manuscript presents the characterization of a secondary cone crusher concave liner made of Hadfield steel used in Chilean mining after crushing copper minerals during all service life. During use, a cone crusher concave liner suffers indentation (cold working) and abrasion. Such combination provides the concave with a layer that constantly renews itself, maintaining a surface highly resistant to abrasive wear. After analysis of results, two hardened surfaces have been found, one product of heat treatment and the other due to deformation during use. The subject matter is important and has a special value considering the application of Hadfield steel. However, there are still things that should be improved, and questions that have to be answered before publication. Therefore, I suggest a mandatory revision of the following points to increase the quality of the paper:

  • Figure 1 is illegible. It should be corrected.
  • The experimental procedure of abrasion resistance measurements should be explained in more detail.
  • The load during hardness measurements should be given in gf (gram-force).

Author Response

(The authors gave the same response as above.)

Reviewer 5 Report

This contribution deals with the investigation of a cone crusher concave liner made of Hadfield steel after service, focusing on wear mechanisms as well as inherent hardening by phase transformation and cold working. Despite generally being well prepared, it suffers from some weaknesses, which should be revised prior to publication. More detailed comments are listed below.  

  1. In terms of language, although generally being well-prepared, a revision is necessary since some typos and mistakes in wording/phrasing can be found throughout the manuscript.
  2. It should be preferred to use SI units (mm instead of inch) or at least to provide both values simultaneously.
  3. Page 2, lines 45 – 48: this statement implies that at a SFE < 18 mJ/m², both martensitic transformation as well as twinning can occur. However, this is not addressed.
  4. In general, both twinning and phase transformation could be investigated by means of EBSD technique. This could strongly improve the contribution.
  5. Figure 1 is confusing since it contains much information, small font, too many different colors. It should be considered to move this to the appendix and present it in terms of multiple equations.
  6. Page 2, lines 62 – 67: there should be a couple of studies in literature dealing with the same topic. They should be cited here and the new / additional output of this contribution needs to be outlined in order to be able to evaluate the impact.
  7. Page 2, line 70: the chemical composition should contain more than 2 elements and it is suggested to present it in terms of a table.
  8. Page 3, lines 85 – 87: this statement can be understood as description of a sequence – if removal is followed by renewal, there should be no wear. However, both mechanisms are likely to occur at the same time with abrasive wear being the dominant factor over a longer period of time.
  9. Page 3, line 91: please provide the Vickers hardness according to the standard (HV0.01 (?)) instead of just stating the test load.
  10. Page 4, line 131: is “duplex size” a common term for such appearance of grains?
  11. Page 4, line 134: it should be considered to provide mass loss in “mg” instead of “g”.
  12. Figure 3: the micrographs show signs of “over-etching” (most probably too long). Furthermore, it seems that there could be some scratches remaining from the grinding and polishing steps.
  13. Page 6, lines 154 – 156: this claim should be supported e.g. by literature reference.
  14. Page 6, line 171: the claim of a low contribution should be supported as well.
  15. Page 6, lines 177 – 179: can martensitic transformation in this case really be called hardening? Epsilon martensite should be not much harder compared to (cold-worked or nano-grained) austenite.
  16. Page 6, it would be worth considering a certain oversize of the part for heat treatment, which could be removed by grinding afterwards. This will solve issues connected to decarburization of surface-near regions.
  17. Page 7, lines 194 – 195: this does not seem to be a surprise since the deformation due to wear / indentation takes place at the surface and only has limited impact on subsurface regions.
  18. Figure 7: these data seem to be derived from testing according to ASTM G65. This should be stated here.
  19. The conclusion could be improved by presenting the main findings in terms of an itemization. Furthermore, please refer to remark no. 6: the impact / outcome of this study in relation to existing literature should be pointed out more distinctly.

Author Response

(The authors gave the same response as above.)

Round 2

Reviewer 1 Report

The authors revised the manuscript.

Author Response

(The authors gave the same response as above.)

Reviewer 2 Report

Dear Authors,

thank you for having improved your paper.

I have carried on some calculation with the simpliefied  Williamson-Hall method obtaining (according to your numbers) a Burger vector of about 3*10-9 m.

Obviously, ..." you state that: "...for this reason, the effects of alloying elements, stacking failures, twins, or others have not been considered."

What is the criteria adopted for the calculation of B ? It is a literature mean value? 

This point is very critical because the B measurement is not easy and it is better to clarify.

Author Response

(The authors gave the same response as above.)

Reviewer 3 Report

much improvement was checked.

Author Response

(The authors gave the same response as above.)
